# Interpreting Morphological Adaptations Associated with Viviparity in the Tsetse Fly *Glossina morsitans* (*Westwood*) by Three-Dimensional Analysis

**DOI:** 10.3390/insects11100651

**Published:** 2020-09-23

**Authors:** Geoffrey M Attardo, Nicole Tam, Dula Parkinson, Lindsey K Mack, Xavier J Zahnle, Joceline Arguellez, Peter Takáč, Anna R Malacrida

**Affiliations:** 1Department of Entomology and Nematology, University of California, Davis, CA 95616, USA; nbtam@ucdavis.edu (N.T.); lkmack@ucdavis.edu (L.K.M.); xjzahnle@ucdavis.edu (X.J.Z.); jarguellez@ucdavis.edu (J.A.); 2Lawrence Berkeley National Lab., Berkeley, CA 94720, USA; dyparkinson@lbl.gov; 3Department of Animal Systematics, Institute of Zoology, Slovak Academy of Sciences, 84506 Bratislava, Slovakia; peter.takac@savba.sk; 4Scientica, Ltd., 831 06 Bratislava, Slovakia; 5Department of Biology and Biotechnology, University of Pavia, 27100 Pavia, Italy; malacrid@unipv.it

**Keywords:** MicroCT, *Glossina*, tsetse, reproduction, morphology, computed tomography, viviparity, trypanosomiasis

## Abstract

**Simple Summary:**

Tsetse flies, the sole transmitters of African Sleeping Sickness parasites, have a unique reproductive biology. They only develop one offspring at a time, they carry that offspring in their uterus for its entire immature development and provide nourishment for that offspring via milk-like secretions. This specialized reproductive biology has required dramatic modifications to the morphology of the reproductive organs in these and related flies. Here, we use phase contrast micro-Computed Tomography (Micro-CT) to visualize these adaptations in three dimensions for the first time. These adaptations include cuticular modifications allowing increased abdominal volume, expanded abdominal and uterine musculature, reduced egg development capacity, structural features of the male seminal secretions and detailed visualization of the gland responsible for synthesis and secretion of “milk” to feed intrauterine larvae. The ability to examine these tissues within the context of the rest of the organ systems in the fly provides new functional insights into how these changes have facilitated the evolution of the mating and reproductive biology of these flies.

**Abstract:**

Tsetse flies (genus *Glossina*), the sole vectors of African trypanosomiasis, are distinct from most other insects, due to dramatic morphological and physiological adaptations required to support their unique biology. These adaptations are driven by demands associated with obligate hematophagy and viviparous reproduction. Obligate viviparity entails intrauterine larval development and the provision of maternal nutrients for the developing larvae. The reduced reproductive capacity/rate associated with this biology results in increased inter- and intra-sexual competition. Here, we use phase contrast microcomputed tomography (pcMicroCT) to analyze morphological adaptations associated with viviparous biology. These include (1) modifications facilitating abdominal distention required during blood feeding and pregnancy, (2) abdominal and uterine musculature adaptations for gestation and parturition of developed larvae, (3) reduced ovarian structure and capacity, (4) structural features of the male-derived spermatophore optimizing semen/sperm delivery and inhibition of insemination by competing males and (5) structural features of the milk gland facilitating nutrient incorporation and transfer into the uterus. Three-dimensional analysis of these features provides unprecedented opportunities for examination and discovery of internal morphological features not possible with traditional microscopy techniques and provides new opportunities for comparative morphological analyses over time and between species.

## 1. Introduction

Tsetse flies (genus *Glossina* (Wiedemann 1830)) are the sole vectors of African trypanosomes, the causative agents of Sleeping Sickness in humans and Nagana in animals. These are fatal diseases predominantly affecting marginalized populations in sub-Saharan Africa which cause severe health and economic impacts in affected countries [1]. Vector control methods are a key component of trypanosomiasis control as tsetse flies have a low reproductive capacity relative to other insects [2]. This is due to specializations in tsetse’s reproductive biology that result in the production of a small number of offspring over a long period of time. Tsetse females reproduce by obligate viviparity which is defined by intrauterine larval development and provision of nourishment by the mother for the duration of development [3,4,5]. Viviparous reproduction is slow relative to oviparous methods with only a single larva produced every 9–10 days. This restricts the number of progeny per female to 8–10 over the duration of her lifespan.

Mating and transfer of seminal fluid into the uterus stimulates behavioral and physiological changes in female tsetse flies. Within 48 h of mating, females become refractory to additional mating attempts by other males, start taking larger blood meals and begin to ovulate. This physiological response is exploited for control purposes via use of the sterile insect technique [6,7]. However, in *Glossina*, little is known regarding the biology underlying the post-mating response. How the post-mating response is activated by mating stimuli in *Glossina* and what happens to internal female morphology in response to these stimuli remains unknown. The reproductive tissues of those in the Hippoboscoidea superfamily, which includes tsetse flies, are highly derived relative to that of other dipterans. The most dramatic modifications include reduced ovarian capacity [8], enhanced uterine musculature [9] and modification of the female accessory gland into a tubular ramified milk-producing gland [10,11,12].

Previous studies analyzing post-mating changes in the reproductive tissues of female *Drosophila melanogaster* revealed that mating induces morphological changes in female reproductive tissues during the post-mating transition [13]. These changes include looping/unlooping of the uterus and oviduct, repositioning of reproductive tract within the abdomen and mating-induced tissue damage resulting from the male reproductive organs. It is hypothesized that the tissue damage may allow male seminal secretions access to the hemocoel of the female. The *D. melanogaster* post-mating response also includes changes in the oviduct which provides a conduit for transfer of oocytes from the ovaries into the uterus. Prior to mating, the oviduct lacks a lumen and is a solid cylindrical tube of muscle and epithelial cells. Following mating, the oviduct undergoes a developmental program resulting in the formation and opening of the duct lumen facilitating passage of oocytes from the ovaries into the uterus [14].

In this work, the abdominal tissues from a whole freshly mated female tsetse fly *Glossina morsitans* (Westwood 1851) (*G. morsitans*) are imaged via phased contrast MicroCT (pcMicroCT) to evaluate the capacity of this technique to perform three-dimensional (3D) analyses of tsetse reproductive morphology. Analysis of the resulting data has provided detailed 3D visualizations of external and internal morphological features and provides new insights into the functional roles of structural features of *Glossina* reproductive organs.

## 2. Materials and Methods

### 2.1. Biological Materials

Tsetse flies (*Glossina morsitans*) utilized in this analysis were obtained as pupae from the colony maintained at the Institute of Zoology at the Slovak Academy of Sciences in Bratislava, Slovakia. Flies were reared in the Tupper Hall arthropod containment level 2 insectary in the UC Davis School of Veterinary Medicine. Flies are maintained in an environmental chamber at 25 ℃ and 75% relative humidity with a 12:12 light/dark photoperiod. Flies receive defibrinated bovine blood via an artificial feeding system Mondays, Wednesdays and Fridays, as described [15]. Sterile defibrinated bovine blood for feeding is obtained from Hemostat Laboratories (Dixon, CA, USA).

### 2.2. Sample Collection

Tsetse pupae were placed into an eclosion cage and monitored for teneral females daily. Teneral flies were anesthetized on ice and sorted into cages by sex. At five days post-eclosion, individual virgin females were combined with males in mating cages. Cages were observed for mate pairing. If pairing was not observed within 10 min, the male was removed from the cage and a new male introduced. Tsetse flies require at least one hour of pairing for completion of the transfer and formation of the spermatophore [16]. Mate pairs lasting for less than 60 min were removed from the sample pool. Within an hour of mating completion, females were anesthetized on ice for sample preparation.

### 2.3. Sample Preparation, Fixation and Staining

Flies were chilled on ice to anesthetize them followed by removal of the legs and wings to allow the fixative to permeate into the haemocoel. The fly was then placed into Bouin’s fixative‘ (Millipore/Sigma, St. Louis, MO, USA) solution (acetic acid 5%, formaldehyde 9% and picric acid 0.9%) and incubated overnight at room temperature. Following fixation, flies were dehydrated using a graded series of ethanol washes (10%, 30%, 50%, 70% and 95%). Each wash was performed for 1 h at room temperature. Flies were then stained in 1% iodine in 100% ethanol for 24 h. After staining, flies were washed 3 times in 100% EtOH (Millipore/Sigma, St. Louis, MO, USA) for 30 min per wash.

### 2.4. Phase Contrast Micro-Computed Tomography

During imaging, samples must remain in a fixed position with no movement during the scanning process to ensure proper alignment of the image stack. In preparation for imaging, fixed and stained flies were transferred into a 1.5 mL Eppendorf tube containing unscented Purell hand sanitizer (Gojo Industries, Akron, OH, USA). The specimen was gently pushed down to the bottom of the sample tube using a pipette tip. To ensure samples remained immobilized during scanning, the bottom of another 1.5 mL Eppendorf tube was cut off and pushed into the sample tube for use as a wedge. The wedge was gently pushed down into the sample tube until the specimen was secured between the wall of the container and the wedge. Once the specimen was secured, the remaining volume of the sample tube was filled with Purell. The sample tube was modified to attach to the sample holder (chuck) in the MicroCT imaging hutch by hot-gluing a wooden dowel (3 mm diameter, 20 mm length) vertically to the flat surface of the Eppendorf tube cap. The sample was imaged using a monochromatic beam of 20 kilo electron volts (keV). Images were captured through a 4× optique peter lens system and a PCO.edge CMOS detector (PCO-Tech Inc., Wilmington, DE, USA) with a 150 mm sample to scintillator distance. The resulting size of the individual sections was 2560 × 2560 pixels with a resolution of 72 pixels/inch and a 32-bit depth. During the imaging process, 1596 images were captured across 180° of rotation. Image stacks in Tagged Image File Format (TIFF) were produced using Xi-CAM [17] with the gridrec algorithm as implemented in TomoPy [18]. The resulting image stack has reconstructed voxel resolution of 1.6 microns. The raw data stack is available for download from the following link https://datadryad.org/stash/share/yH7IFkgeWtlX07qIp3Qu5DXpt7EtM_lMGv5s7wrjoBg.

### 2.5. Data Processing, Segmentation, Visualization and Analysis

Data analysis and visualization was performed in the Dragonfly software package version 4.0 developed by ORS (Object Research Systems, Montreal, QC H3C 1M4, Canada). Software is available at http://www.theobjects.com/dragonfly. Image volumes were imported into Dragonfly and down-sampled to 16-bit depth at half resolution (1280 × 1280) to improve computer performance during analysis and segmentation. The image volume was cropped to eliminate portions of the volume not occupied by the sample. Upon import, dataset contrast and sharpness were enhanced using the Unsharp image processing function with a kernel size of 7, standard deviation of 3 and unsharp factor of 3. Tissue segmentation and region of interest definitions were performed using a combination of algorithmic and manual methods. Three mated females were scanned for this project. Based on visual analysis of the scans, a single scan was chosen randomly for detailed segmentation and analysis as the quality appeared equivalent between scans. Images were captured and exported from within Dragonfly. Tissue volumes, surface areas and thickness mapping were calculated by measurement of segmented voxels by Dragonfly with each voxel representing 1.6 uM^3^. The associated video was generated using the Dragonfly movie maker function. The video is available for download from https://drive.google.com/file/d/1bHU2A6Fsxb_ZuJkg3gnbfAWXuQmr-TLG/view?usp=sharing.

## 3. Results and Discussion

### 3.1. Abdominal Structural and Cuticular Adaptations for Blood Feeding and Pregnancy

The analysis described here encompasses the last four abdominal segments of a female tsetse fly (*Glossina morsitans*) immediately after copulation (Figure 1A). This region contains the entirety of the reproductive tract as well as tissues from other organ systems, such as the digestive tract, abdominal musculature, respiratory system and the fat body (nutrient storage/metabolism). The presence of these other systems within the scan provides spatial context of how the reproductive tract interacts with other abdominal tissues.

Analysis of the exoskeleton reveals features required to accommodate the massive changes in abdominal volume required during blood feeding and pregnancy. As in most blood-feeding insects, tsetse flies ingest large volumes of blood relative to their body size. After mating, females on average take blood meals weighing between 50 and 60 mg [19]. In addition, pregnant females accommodate fully developed third instar larvae equivalent in mass to themselves. Newly enclosed female flies weigh ~18 mg and just after a blood meal, a pregnant female can weigh over 90 mg prior to water elimination via diuresis [19]. Thus, females are capable of a 5-fold change in mass over the course of a pregnancy cycle. It follows that to accommodate these dynamic changes, the abdomen must undergo a significant increase in volume.

A detailed view of exterior (Figure 1A) and interior (Figure 1B) structural features reveals some of the adaptations facilitating abdominal expansion and elasticity. The dorsal surface of the abdominal cuticle forms pleats (dorsal intersegmental pleats) at the junction of the abdominal segments. The dorsal cuticular plates are sclerotized and covered in setae while the cuticle connecting the plates is unsclerotized and pliable. This connective cuticle folds under the proceeding segment, allowing the dorsal surface of the abdomen to expand and contract along the length of the abdomen. The ventral surface of the abdomen differs from the dorsal surface as the cuticle is soft and pliable, resembling the connective cuticle associated with the dorsal plates. The ventral cuticle is marked by longitudinal folds that, in addition to cuticle pliability, allow it to accordion out and expand to accommodate large changes in volume. The ventral abdominal folds bear functional similarities to the ventral pleats of rorqual whales that feed on krill and schools of small fish. These whales also undergo huge changes in volume during feeding as they swallow volumes of water equivalent to or larger than themselves [20]. The similarities of these features suggest a convergent solution to the problem of accommodating rapid and large changes in volume.

The sagittal section through the abdomen reveals how tightly packed the abdominal organs are in this space (Figure 1B). The posterior volume of the abdomen is dominated by the ovaries, uterus and spermatheca, with the hindgut traveling over the top of the reproductive tract and terminating at the anus. The anterior section of the abdomen is primarily occupied by the midgut which winds throughout the abdomen. The bacteriome region of the midgut which houses the obligate symbiont *Wigglesworthia* is clearly visible and positioned along the center line of the abdomen and is anteriorly adjacent to the reproductive tract. The remaining volume is predominantly occupied by fat body, tracheae and malphigian tubules.

### 3.2. Functional Aspects of G. morsitans Abdominal Musculature

Internal examination of the dorsal segmental pleating reveals that the medial portion of each pleat is connected to the dorsal plate anterior to it via an array of muscles (dorsal intersegmental retractor musculature—DIRM) (Figure 1C,D). Each muscle is angled medially from its anterior origin to its insertion onto the succeeding dorsal plate. The medial most DIRM of the final two segments have paramedian origins but converge to the midline at their insertions. Contraction of DIRM muscles looks to pull the intersegmental pleats in towards the plates anterior to them, which would result in retraction of the dorsal abdominal segments. The angle at which the muscles are oriented suggests that they aid in the birthing process by retraction of the dorsum to assist in dilating the vaginal canal. This musculature appears to function indirectly, relative to the rest of the uterine musculature, as it does not directly connect to the reproductive tract.

On the ventral side, a wide band of muscles (ventral intersegmental retractor musculature—VIRM) spans the third abdominal segment (Figure 1C,D). This band comprises two distinct pairs of longitudinal muscles: one paramedian pair and one lateral pair. Each VIRM is distally continuous with a muscle that inserts onto the anterior face of the uterus (Figure 1D). Of these uterine muscles, the lateral pair (diagonal anterior uterine dilatory musculature—DAUD) extends diagonally from the medial VIRM toward midline of the uterus. The second set of uterine muscles (anterior uterine dilatory muscle—AUD) extend as a pair from the ventral VIRM to insert onto the ventral side of the uterus. This set of muscles anchors the uterus to the ventral abdominal wall providing leverage to aid in dilation of the vaginal opening as well as to squeeze the anterior portion of the uterus to push the intrauterine larva towards the vaginal opening.

### 3.3. The External Uterine Musculature is Optimized for Intrauterine Larval Development and Parturition

The uterus in *G. morsitans* during this stage is compact and covered in an elaborate symmetrical array of muscle pairs that anchor it to adhesion points in the final abdominal segment and the VIRM. The muscle groups were defined by manual annotation of the muscle striation patterns across the surface of the uterus using previously established nomenclature (Figure 2) [9]. The seven uterine muscles can be divided into three groups based on their origins: (1) transverse sulcus separating third and fourth ventral segments—DAUD and AUD, (2) ventral posterior abdominal wall— the posterior uterine constrictor muscle (PUC) and the anterior uterine retractor muscle (AUR) and (3) terminal dorsal abdominal segment—diagonal posterior uterine dilator muscle (DPUD), dorsal uterine dilator muscle (DUD) and the vertical posterior uterine muscle (VPU). Muscle groups 2 and 3 have extensive connections to cuticular invaginations and structures in the terminal abdominal segment.

Among muscles originating at the ventral posterior wall, the posterior uterine dilator musculature (PUC) originates at the midline of the uterus at a point ventral to the vaginal opening. It then runs antero-laterally along the ventral and lateral sides of the uterus, passes deep to the dorsal uterine dilator musculature (DUD) and inserts on the dorso-lateral sides of the uterus midway along its length. Due to being covered by the DUD, the insertion of the PUC is not visible in Figure 2. The PUC may aid in contraction of the uterus by providing lateral compression during parturition. The anterior uterine retractor muscle (AUR) anchors to the surface of two bowl-shaped cuticular structures that jut into the abdominal cavity from the ventral posterior abdominal wall. The AUR musculature pair cross antero-medially deep to the PUC and meet anteriorly along the ventral midline of the uterus. The AUR muscles split dorso-laterally along the anterior face of the uterus and follow the left and right sides of the ovarian shelf, finally inserting onto the postero-lateral corners of the oviductal shelf. The AUR musculature acts like straps anchoring the uterus to the posterior wall of the abdomen and can compress the anterior wall of the uterus towards the abdominal posterior. This would generate force along the antero-posterior axis to maintain uterine structure during embryogenesis/larvigenesis and aid in pushing the intrauterine larva out of the vaginal canal during parturition.

The dorsal posterior region of the uterus is anchored to the last abdominal segment by three musculature groups, the diagonal posterior uterine dilator musculature (DPUD), the dorsal uterine dilator musculature (DUD) and the vertical posterior uterine muscle (VPU). The DPUD originates on the internal dorsal surface of the last abdominal segment. It then extends ventro-medially as two bundles of muscle at a 45° angle and inserts on the posterior end of the uterus, deep to the AUR and PUC. The DUD extends anteriorly as a horizonal sheet from its origins along the posterior wall of the terminal abdominal segment. It runs parallel along the dorsal surface of the uterus inserting at the base of or underneath the ovarian shelf. The attachments made by this muscle group look to maintain close association of the posterior dorsal surface of the uterus with the last abdominal segment. The final muscle group, the VPU, originates on the inner dorsal cuticle, like the DPUD. However, the VPU origins are more medial such that the VPU extends ventrally to insert onto the dorsal surface of the vaginal opening. The muscle bundles from the VPU extend through the DUD musculature which flows around these muscles and the lateral parts of the vaginal opening prior to connection to the cuticle. The vertical positioning of the VPU suggests it also acts to secure the uterus to the last abdominal segment as well as to facilitate the opening and closing of the vaginal canal.

This analysis has revealed the basic components of the uterine musculature and additional work is in progress to compare the conformational changes these structures undergo in response to mating stimuli, ovulation and the pregnancy cycle to better understand their functional roles. The other families within the Hippoboscoidea superfamily (Hippoboscidae (Ked flies), Nycteribiidae (Bat flies) and Streblidae (Bat flies)) are all obligately hematophagous and undergo obligate viviparity. This suggests that members of these families likely have similar reproductive adaptations, but may also have species- or family-specific specializations. Comparative analysis of *G. morsitans* uterine morphology with that of other Hippoboscoidea and of oviparous members of the brachyceran Diptera, such as *Drosophila* and members of the Oestroidea and Muscoidea superfamilies, can inform as to the derivation of these adaptations.

### 3.4. Ovarian Reduction and Oviduct Folding Reduce Constraints on Blood Meal Volume and Larval Size

Another major adaptation to *Glossina* reproductive physiology, relative to oviparous Diptera, is the reduction in ovarian capacity. The ovaries of most oviparous Diptera contain dozens of ovarioles with some or all containing vitellogenic oocytes at the same time [21]. In *Glossina*, each ovary contains two ovarioles with only one mature oocyte completing development per gonotrophic cycle [8,22,23] (Figure 3A,B). Cross-section through the developing ovarian follicles shows the fully developed primary follicle with the tertiary ovarian follicle in close association inside the right ovary (Figure 3C). In the primary follicle, the oocyte has filled with yolk protein crystals and lipids and the nurse cells have emptied into the oocyte and collapsed. This oocyte has completed most of its development and is almost ready for ovulation into the uterus.

Tsetse flies require mating-associated stimuli to undergo ovulation [24,25,26]. The fly visualized in this analysis is five days post-eclosion and had just completed mating. The first ovulation usually occurs around day nine or ten post-eclosion. The tertiary follicle contains nurse cells and is surrounded by follicular epithelial cells; however, it lacks a defined oocyte or any yolk deposition. The left ovary contains the secondary and quaternary ovarian follicles. The secondary follicle contains a differentiated oocyte with large nurse cells surrounded by follicular epithelial cells and a defined oocyte containing a small amount of yolk. This oocyte remains in stasis until the primary oocyte has been ovulated into the uterus. After ovulation of the primary oocyte, the secondary follicle will begin rapid accumulation of yolk proteins and nutrients (vitellogenesis) in preparation for the next gonotrophic cycle. The quaternary ovarian follicle (not visible) is early in development, containing nurse cell primordia and a follicular epithelium. During the initial gonotrophic cycle, the process of oogenesis always begins with one of the follicles in the right ovary [23]. Follicular development cycles through the four ovarioles in sequence, shifting between ovaries each gonotrophic cycle.

During ovulation, the fully developed oocyte moves out of the ovary, through the oviduct and into the uterus. In *G. morstians*, the oviduct and its associated tissues form a structure called the oviductal shelf which is a sleeve of tissue connecting the ovaries to the uterus (Figure 3D). The oviductal shelf folds over on itself and occupies most of the dorsal surface of the uterus. The lack of connective musculature on the dorsal anterior portion of the uterus is likely due to the spatial constraints associated with these structures. The folded conformation results in the oviduct forming a hairpin turn prior to its opening into the uterus. The compacted structure of the oviductal shelf in addition to reduced ovarian capacity optimizes the space occupied by the reproductive tract within the abdominal cavity. These optimizations allow for larger blood meal volumes and increased nutrient storage capacity by the fat body between larvigenic cycles. During ovulation, the oviduct and oviductal shelf likely must expand and straighten to allow the oocyte to pass into the uterus.

The sagittal section of the uterus reveals that the anterior wall of the uterus also appears folded upon itself and is likely capable of expanding to accommodate the large volume required by the developing intrauterine larvae (Figure 3D). In addition, the tissue constituting the anterior and posterior uterine walls shows patterning which may represent additional folding that provides elasticity and flexibility to the uterine walls. The entire reproductive tract is optimized to conserve space while retaining the capacity to expand and occupy the majority of the abdominal volume during intrauterine larval development.

### 3.5. The Spermatophore Facilitates Sperm Storage within the Spermatheca and Acts as a Physical Barrier to Insemination by Competing Males

In *Glossina*, physical and chemical mating stimuli are required to initiate ovulation and activate other post-mating changes including initiation of sexual refractoriness, accelerated oocyte development, increased host seeking/blood feeding and ingestion of increased blood meal volumes [16,22,24,27,28]. Mating involves the transfer of a large volume of male accessory gland-derived proteins and biochemicals which form a well-defined structure called a spermatophore in the uterus of the female [29,30,31,32]. The center of this structure is hollow and contains sperm that are stored in the spermathecal organ of the female. The outer wall of this structure is composed of two layers with differing ultrastructural characteristics and has an opening at the dorsal anterior [33]. Visualization of the spermatophore reveals that the outer walls are molded to the interior of the uterus and that it occupies a significant volume of intrauterine space (Figure 4B).

The sperm storage organs, the spermatheca, and associated spermathecal ducts lie on the dorsal surface of the uterus and ovaries. The spermathecal ducts connect to the uterus as a paired tubule and share a common opening on the dorsal interior wall of the uterus (Figure 4B). From there, the paired ducts migrate through the uterine wall and emerge from under the oviductal shelf, where they curve to the anterior and then split into left and right tubules flanking the shelf. The tubules then rejoin dorsal to the ovaries as open into the two spermathecal capsules (Figure 4). The inside of the spermathecal capsule is lined with chitin and surrounded by epithelial cells which produce and secrete proteins and nutrients required to maintain the viability of stored sperm for the duration of the female’s lifespan (Figure 4E).

The lacuna of the spermatophore opens at the entrance to the spermathecal ducts in the uterus and creates a seal against the entryway to the spermathecal ducts. This conformation guides the sperm to the opening of the ducts where they travel up to the spermathecal capsule for storage. The spermatophore also blocks access to the uterus by other males, with the posterior wall inhibiting entry of seminal components via the vaginal canal. Females have been observed containing two spermatophores [31]. However, the spermatophore resulting from the second mating attempt is usually stuck to the back of the first which results in a physical barrier preventing sperm from exiting the lacuna of the spermatophore. Even if the sperm were able to exit the spermatophore, it is unlikely they would be able to navigate around the first spermatophore to access the spermathecal ducts.

Among the Diptera, the use of spermatophores by males is infrequent [34]. However, in tsetse flies inter- and intra-sexual competition is intense due to the low reproductive rate of females. There is evidence for cryptic selection by females as mating with substandard males can result in poor sperm uptake and continued receptivity to other males [35]. Comparative analysis of male seminal protein genes between six *Glossina* species revealed them to be the most rapidly evolving genes in the genome, with observed differences in gene number and sequence variability [36]. The use of the spermatophore by males may increase the probability that a mating attempt is successful in the face of female selective pressures. The structural advantages of the spermatophore for males are that it is difficult for the female to expel, it guides the sperm to their destination and inhibits insemination by sperm from competing males. Females normally dissolve the spermatophore and expel its remnants ~24 h post mating and become refractory to further mating ~48 h post-mating [29].

### 3.6. The Milk Gland Organ Maximizes Nutrient Incorporation via Extensive Ramification for Increased Surface Area and Intimate Contact with Fat Storage Tissues

The final component of the reproductive tract is the milk gland. The milk gland is a modified female accessory gland which is responsible for the synthesis and secretion of a protein- and lipid-rich milk-like secretion. The milk gland is a dynamic organ which undergoes cyclical changes in volume throughout the reproductive cycle in correlation with milk production activity [10,12]. In *Glossina austeni*, it is estimated that the milk gland produces 25 mg of milk secretion per gonotrophic cycle, which is equivalent to the weight of an unfed adult female [10]. To accomplish this feat, the gland must incorporate large amounts of stored lipids and free amino acids from dietary sources [37,38]. The milk gland does not seem capable of protein uptake from the hemolymph and instead imports amino acids which are utilized by the massive arrays of rough endoplasmic reticulum for milk protein synthesis [10,12]. The milk gland also harbors the extracellular form of the *Glossina* obligate symbiont *Wigglesworthia glossinidius*. The bacteria live in the lumen of the gland and are transferred to the intrauterine larva during lactation [11,39].

Prior analysis of the milk gland has been primarily by two-dimensional (2D) microscopic analysis with hand-drawn interpretations of the 3D structure of this complex organ. The analysis of the abdominal volume from the pcMicroCT scan has provided a detailed virtual representation of the milk gland which clearly demonstrates the intricate nature of its branching structure (Figure 5A). The gland consists of three regions, the distal milk gland, the proximal milk gland and the common collection duct. The milk gland is connected to the dorsal surface of the uterus via the common collection duct. This duct connects to the uterus under the oviductal shelf just posterior to the opening of the spermathecal duct (Figure 5A). The duct extends through the musculature of the uterine wall and travels along the right side of the oviductal shelf, where it crosses dorsally over the right spermathecal duct. The common collection duct is formed of two cuticle-lined tubes lacking in secretory cells that are bundled together by a spiraling musculature, which likely regulates the flow of the milk secretions [10,12]. At the point where the ovaries begin, the common collection duct branches into four thick tubules. Two of these cross dorsally over of the spermathecal ducts and spermatheca to expand into the left side of the abdomen, while the other two proximal tubules extend to the right. The proximal milk gland lacks the musculature that wraps around the common collecting duct and contains secretory cells that contribute to milk production. The proximal milk gland transitions into the distal milk gland, which is the dominant part of the milk gland in terms of space occupied in the abdomen. The distal milk gland primarily consists of secretory cells and epithelial cells that line the interior of the gland lumen. The distal milk gland is extensively ramified and extends throughout the volume of the abdomen, with most of it found within the last three segments of the abdomen. Quantitative analysis of the thickness of the gland shows that it ranges from 10 to 12 µm at the tips of the distal milk gland to 42 µm in diameter along the length of the common collecting duct (Figure 5B). The proximal gland is of intermediate diameter ranging from between 15 and 30 µm with width decreasing as it transitions into the distal milk gland. The tubules of the milk gland are known to dynamically expand in diameter over the course of the pregnancy cycle [12].

An inherent aspect of the tubular structure of the milk gland is that it has a very high surface area to volume ratio. Analysis of the milk gland in the context of the surrounding abdominal tissues demonstrates that it is intimately associated with fat body cells which store large volumes of lipids for milk production (Figure 5C). The large surface area of the gland facilitates rapid transfer of stored lipids during lactation. Another interesting observation is that some of the milk gland tubules are proximal to the bacteriome. The bacteriome is constituted of bacteriocyte cells that house intracellular *Wigglesworthia* and forms a horseshoe-like structure that expands into the anterior midgut [40]. The mechanism by which *Wigglesworthia* invade the milk gland remains unknown and the bacteria are not found in any other tissue in the fly. Higher resolution scans of the bacteriome could provide additional information on potential structural associations between these tissues and whether *Wigglesworthia* is capable of movement between these two organs.

### 3.7. Volumetric Relationships and Functional Implications

The nature of this data facilitates observation of the spatial relationships between tissues with diverse functions working synergistically to achieve the overarching goal of reproduction. Comparison of the respective volume to surface area relationships reveals structural optimizations to reproductive tissues associated with functional roles (Figure 5D). The ovaries and uterus comprise 7.33% of the abdominal volume but have a relatively low surface area to volume ratio, revealing its spatial efficiency. In contrast, the milk gland and muscle tissues only comprise ~1% of the abdominal volume yet have 4 and 2.5 times the surface area to volume ratio, respectively. This difference reflects the functional necessities of these tissues. The milk gland requires abundant contact with surrounding tissues to optimize nutrient transfer. The higher surface area of the muscle tissue results from the banding across the dorsal and ventral abdominal surfaces. The large surface area allows the musculature to distribute force evenly across the interior of the abdomen wall. An overview of the reproductive tract has also been provided as an animated video which provides additional views of these organs (Appendix A).

## 4. Conclusions

The capabilities of pcMicroCT provide new opportunities for the study of soft tissue morphology. The non-invasive nature of this technique allows the observation of delicate relationships between internal features without the disruption caused by dissection and sectioning techniques. Once preserved, samples can be stored for years and multiple scans can be taken of the same sample at different resolutions or in different regions to highlight other morphological aspects. The ability to view external and internal structures from any angle and then to isolate and view specific features in three dimensions provides additional context, aids in viewer comprehension and facilitates deeper analysis of the data. Another tremendous benefit to this technique is that scanned volumes can be made publicly available similarly to datasets from other high-throughput technologies. Users can download these volumes for independent analysis of features that were not a focus of the original study. Finally, the nature of this data allows export of it to new 3D visualization technologies such as virtual reality applications. With the aid of a virtual reality headset, the user can explore the physiology interactively in 3D at an otherwise impossible scale. The addition of spatial cues with rich visuals leverages the natural capabilities of the human brain to rapidly interpret and remember information provided in this format [41,42,43].

Still, there are significant limitations associated with this technology. The availability of facilities capable of pcMicroCT are, for the most part, constrained to those associated with a synchrotron. The scans used in this study were performed at the Lawrence Berkeley National Laboratory Advanced Light Source (ALS) on Beamline 8.3.2. The ALS is funded by the United States Department of Energy. Access to beamtime is provided to users on a grant-based system. However, new lab-based MicroCT technologies are available or are in development from companies such as Zeiss and Bruker that have or will integrate phase contrast capabilities.

A powerful feature of traditional microscopy is the ability to visualize gene expression patterns or protein localization via techniques such as in situ and immunohistochemical staining. Protocols for performing these types of analyses via microCT are in their infancy and are complicated by accessibility and even diffusion of staining and washing reagents in a whole mounted specimen. In addition, traditional staining reagents do not absorb high-energy X-rays, making them invisible to this imaging technique. However, metallic staining reagents coupled to horse radish peroxidase reactive substrates show promise as a way to address these issues [44].

Finally, efficient analysis of the large volumes of data generated by these scans is hindered by the time required for accurate annotation. Segmentation of soft tissues with similar X-ray absorbances is difficult if not impossible for automated algorithms. Phase contrast helps with this but may still be inadequate for basic contrast-based segmentation programs. The tissue segmentations performed for this volume were done mostly by hand with aid from contrast-based analysis tools and predictive algorithms, which is why this work was limited to analysis of a single volume. Recent comparative MicroCT-based work was performed comparing the larval digestive tract across three larval instars in the sheep nasal bot fly species (*Oestrus ovis* L.). In addition, they made comparisons with two other species *Cephenemyia stimulator* (Clark) and *Hypoderma actaeon* (Brauer). This study discovered significant differences in the morphology of the digestive tract over instars and between species. highlighting the potential for comparative analysis using this technique [45]. The authors there noted that the time investment required to perform a detailed segmentation of scanned volumes is a limiting factor. The results of these scans often require a human eye to correctly determine boundaries between adjoining tissues. However, new methods based on cutting-edge artificial intelligence techniques such as machine learning and deep learning are in development. Future work using this technology will utilize the segmentations performed here to train these algorithms to increase analysis throughput and facilitate comparative analyses. The software used for this analysis Dragonfly version 4.1 (Object Research Systems, Montreal, QC, H3C 1M4, Canada).) implements deep learning algorithms that utilize manually segmented datasets to train the algorithm which can then be used to automatically segment raw datasets with minimal human intervention. These methods are still in early stages, but will develop rapidly as more data becomes available, computing power increases and more users develop training sets [46]. This technique was recently used to visualize the interactions between Cordyceps fungus interactions with the brain and musculature of carpenter ants [47]. The segmentations generated by this analysis will be used as training data with which to segment additional datasets. The benefits of MicroCT studies for physiological and morphological analyses are clear and it is a powerful tool that will provide a novel way to approach morphological and physiological studies of vector biology and vector parasite interactions.

## Figures and Tables

**Figure 1 insects-11-00651-f001:**
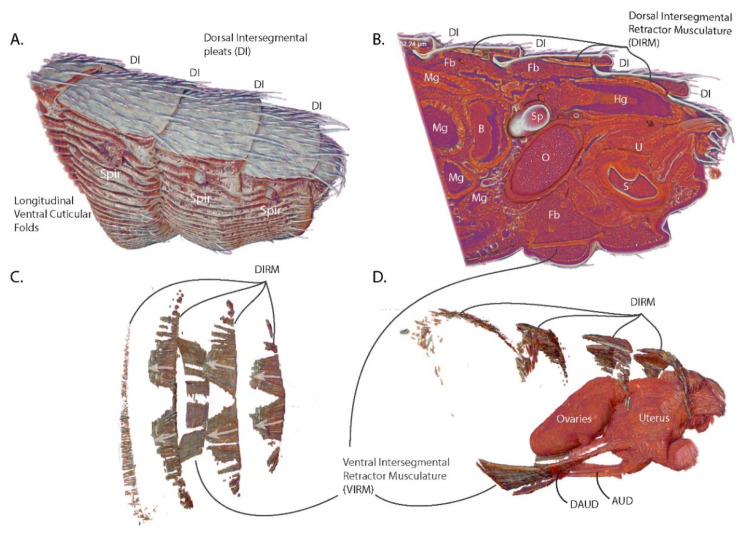
Overview of the scanned abdominal volume of the last four abdominal segments of a female tsetse fly (Glossina morsitans, (Westwood 1851) (G. morsitans)) 2 h post-mating. (**A**) Side view; (**B**) Sagittal Section; (**C**) Dorsal View of the abdominal musculature; (**D**) Lateral view of the abdominal musculature with uterus and ovaries for context. Abbreviations: DI—dorsal intersegmental pleats, Fb—fat body, Mg—midgut, B—bacteriome, Sp—spermatheca, Hg—hindgut, O—ovaries, U—uterus, S—spermatophore, Spir—spiracle, DAUD—diagonal anterior uterine dilatory musculature, AUD—anterior uterine dilatory musculature, DIRM—dorsal intersegmental retractor musculature, VIRM—ventral intersegmental retractor musculature.

**Figure 2 insects-11-00651-f002:**
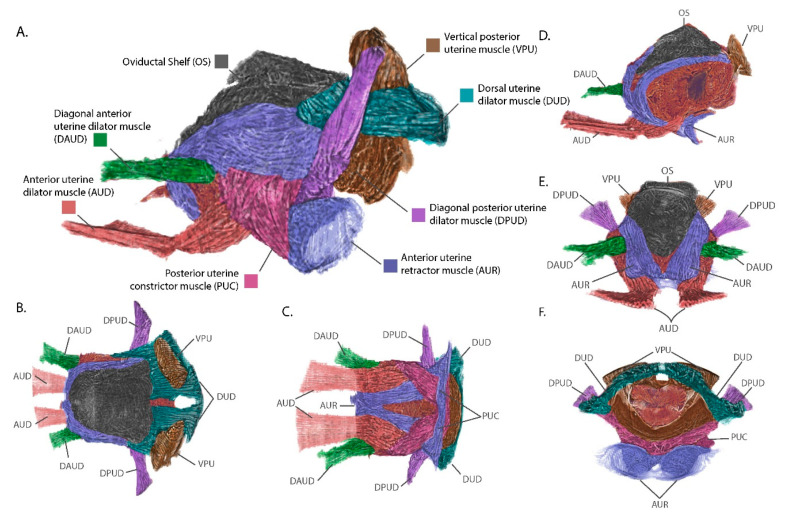
Views of the uterus and associated connective tissues with annotated musculature groups. (**A**) Lateral view; (**B**) Dorsal view; (**C**) Ventral view; (**D**) Orthogonal view with sagittal section. (**E**) Anterior view. (**F**) Posterior view. Muscular group colorations and acronyms are defined in the figure legend.

**Figure 3 insects-11-00651-f003:**
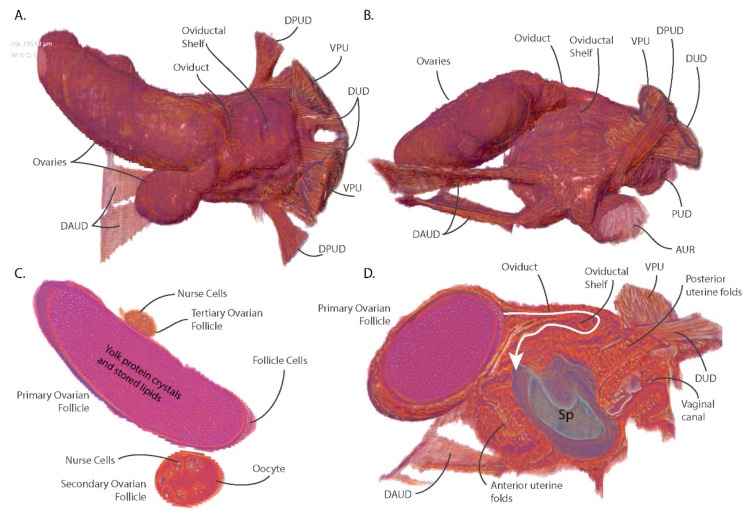
Views of the ovarian and uterine tissues highlighting ovarian structure, oocyte development and ovulation mechanics. (**A**) Dorsal view; (**B**) Lateral View; (**C**) Coronal section of ovarian follicles; (**D**) Sagittal section through the ovaries and uterus highlighting the path of the oviduct and the spermatophore inside the uterus. Abbreviations: DPUD—diagonal posterior uterine dilatory musculature, VPU—vertical posterior uterine musculature, DUD—dorsal uterine dilator musculature, DAUD—diagonal anterior uterine dilator musculature, AUR—anterior uterine retractor musculature.

**Figure 4 insects-11-00651-f004:**
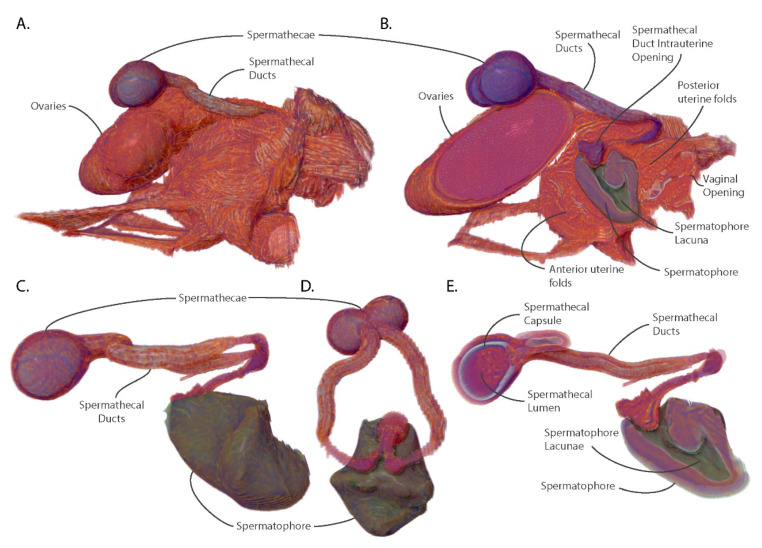
Views of the spermathecae, spermathecal ducts and spermatophore with uterus and ovaries for context. (**A**) Lateral view of uterus, ovaries and spermatheca; (**B**) Sagittal section of the uterus, ovaries, spermatheca and spermatophore; (**C**) Lateral view of isolated spermatheca and spermatophore; (**D**) Anterior view of isolated spermatheca and spermatophore; (**E**) Sagittal section of spermathecae and spermatophore.

**Figure 5 insects-11-00651-f005:**
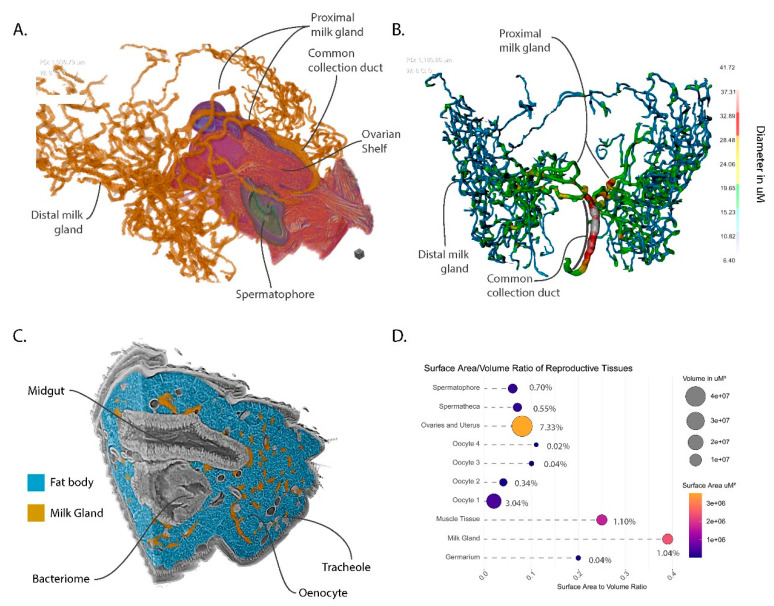
Views of milk gland morphology and structural features with other reproductive and abdominal features for context. (**A**) Orthogonal view with sagittal section through uterus, ovaries and spermatheca; (**B**) Dorsal view of a thickness mesh of the milk gland; (**C**) Orthogonal view of the abdominal volume with milk gland and fat body denoted by color; (**D**) Quantification and visualization of relative volumes, surface areas, surface area to volume ratios and percentage to total abdominal volume within the scan. Circle sizes represent volume in µm^3^, circle color represents surface area in µm^2^, percentages represent proportion of the total volume of the abdominal tissue included within the scan.

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
