# Peer review of "Interpreting Morphological Adaptations Associated with Viviparity in the Tsetse Fly Glossina morsitans (Westwood) by Three-Dimensional Analysis"

_insects, 2020, doi:10.3390/insects11100651_

Round 1

Reviewer 1 Report

Manuscript title: Interpreting morphological adaptations associated with viviparity in the Tsetse fly (Glossina morsitans) by three-dimensional analysis

General comments: This is very worthwhile, relevant and interesting study providing new valuable insights into the morphological adaptations of a medically important species. It also shows the potential of novel imaging methods for medical entomology research. Several results, like the intimate association of the milk gland with the fat body cells, are particularly interesting. I have some comments and suggestions that the authors may want to consider should they decide to revise their manuscript:

- All the genus and species names in the manuscript should be written in italics. Moreover, it would be advisable to include the taxonomic authority for each of them the first time each genus or species is mentioned.

- Lines 62-65: I miss a brief mention to other Hippoboscoidea families, as they also show adenotrophic viviparity. As it reads now, it suggests that this is a unique feature of Glossinidae among Diptera.

- Line 67: The clarification of “D. melanogaster” as an abbreviation of Drosophila melanogaster is not needed. The genus can be directly abbreviated the next time the species is mentioned (line 72) as it is a standard rule. See also the comment above on the use of italics for genus and species names.

- Line 70: “Its hypothesized” should be “It is hypothesized”.

- Line 102: It seems that only one specimen was scanned, so how was it selected from the group of flies that had been fixed? Just at random? Please, clarify this. It might also be worthwhile to add one or two “self-critique” sentences in the discussion regarding the use of only one specimen for the study. Such a small sample size does not affect the relevance of the study and of course the access to scanning facilities can indeed be a limitation (as suggested by the authors in lines 444-448, but it might be good to simply acknowledge that future studies could be improved with increased sample sizes.

- Lines 176-180: The authors provide an interesting comparison with rorqual whales regarding the morphological adaptations to accommodate large changes in volume after feeding, but what happens with other hematophagous insects (and more particularly other flies) where the abdomen also expands considerably after a blood meal? Is it a similar mechanism?

- Line 188: It might be better to use “tracheae” or “tracheal system” instead of the singular form “trachea”.

- Line 189: In this case, the genus or species name should not be in italics, as the rest of the subheading is in italics. Moreover, I think that the authors should be more consistent and use the species name (G. morsitans) and not just the genus name here and hereinafter in the manuscript, as their results are based on a specimen from that species and therefore cannot be directly applied to the entire genus (even if major differences in other Glossina species are unlikely).

- Lines 258-259: See my comment above on the convenience of a brief explanation of the position of Glossinidae flies within the Hippoboscoidea and the common reproductive features. Many potential readers might be unaware of that so a simple clarification would not bother.

- Line 260: A comparison with Drosophila would be of course interesting and a logical option given the model organism status of fruit flies, but perhaps it would be more interesting to compare with more closely related calyptrate flies (Musca but also other Muscoidea and Oestroidea flies).

- Lines 472-475: The unparalleled opportunities provided by microCT to morphological studies are enormous and this paper is a very good example, but I like that the authors also emphasize the specific opportunities for medical and veterinary entomology studies. In this sense, a potentially interesting microCT study related to tsetse flies reproduction and development could be describing the intrauterine larval development using this non-invasive technique. A very recent study has described the morphology of the excretory and digestive systems through the larval instars of bot flies using microCT (Major differences in the larval anatomy of the digestive and excretory systems of three Oestridae species revealed by micro‐CT; https://doi.org/10.1111/mve.12476), and visualizing the development of those systems would be particularly interesting in the larvae of tsetse flies.

Author Response

We sincerely thank the reviewers for the time and consideration they have given to this manuscript. We have implemented changes as suggested by the reviewers. The changes address the comments point by point below.

Reviewer 1:

Comments and Suggestions for Authors

Manuscript title: Interpreting morphological adaptations associated with viviparity in the Tsetse fly (Glossina morsitans) by three-dimensional analysis

General comments: This is very worthwhile, relevant and interesting study providing new valuable insights into the morphological adaptations of a medically important species. It also shows the potential of novel imaging methods for medical entomology research. Several results, like the intimate association of the milk gland with the fat body cells, are particularly interesting. I have some comments and suggestions that the authors may want to consider should they decide to revise their manuscript:

- All the genus and species names in the manuscript should be written in italics. Moreover, it would be advisable to include the taxonomic authority for each of them the first time each genus or species is mentioned.

All genus and species names have been italicized and updated with the appropriate taxonomic authority

- Lines 62-65: I miss a brief mention to other Hippoboscoidea families, as they also show adenotrophic viviparity. As it reads now, it suggests that this is a unique feature of Glossinidae among Diptera.

This line has been revised to read “The reproductive tissues of those in the tsetse fliesHippoboscoidea superfamily,which includes tsetse flies, are highly derived relative to that of other dipterans.”

- Line 67: The clarification of “D. melanogaster” as an abbreviation of Drosophila melanogaster is not needed. The genus can be directly abbreviated the next time the species is mentioned (line 72) as it is a standard rule. See also the comment above on the use of italics for genus and species names.

The reference to Drosophila melanogaster has been revised to match standard formatting rules.

- Line 70: “Its hypothesized” should be “It is hypothesized”.

Corrrected

- Line 102: It seems that only one specimen was scanned, so how was it selected from the group of flies that had been fixed? Just at random? Please, clarify this. It might also be worthwhile to add one or two “self-critique” sentences in the discussion regarding the use of only one specimen for the study. Such a small sample size does not affect the relevance of the study and of course the access to scanning facilities can indeed be a limitation (as suggested by the authors in lines 444-448, but it might be good to simply acknowledge that future studies could be improved with increased sample sizes. See highlight in manuscript:

We limited this study to a single specimen due to the time constrains required for manual annotation. The three samples we scanned were of equivalent quality, so one was chosen to be representative. We are working on techniques leveraging this annotated dataset to train a deep learning algorithm in increase the speed of the analyses for future work. We have added this line to the materials and methods section (lines 138-140).

“Three mated females were scanned for this project. Based on visual analysis of the scans a single scan was chosen randomly for detailed segmentation and analysis as the quality appeared equivalent between scans.”

We have also added the following to the discussion section (lines 469-475).

“The tissue segmentations performed for this volume were done mostly by hand with aid from contrast-based analysis tools and predictive algorithms, which is why this work was limited to analysis of a single volume. The results often require a human eye to correctly determine boundaries between adjoining tissues. However, new methods based on cutting edge artificial intelligence techniques such as machine learning and deep learning are in development. Future work will utilize the segmentations performed here to train these algorithms to increase analysis throughput and facilitate comparative analyses.”

- Lines 176-180: The authors provide an interesting comparison with rorqual whales regarding the morphological adaptations to accommodate large changes in volume after feeding, but what happens with other hematophagous insects (and more particularly other flies) where the abdomen also expands considerably after a blood meal? Is it a similar mechanism?

We are not sure about how other hematophagous insects facilitate their abdominal expansion. This is something we are investigating using this technology to perform comparative analyses.

- Line 188: It might be better to use “tracheae” or “tracheal system” instead of the singular form “trachea”.

“Trachea” has been replaced with “tracheae”

- Line 189: In this case, the genus or species name should not be in italics, as the rest of the subheading is in italics. Moreover, I think that the authors should be more consistent and use the species name (G. morsitans) and not just the genus name here and hereinafter in the manuscript, as their results are based on a specimen from that species and therefore cannot be directly applied to the entire genus (even if major differences in other Glossina species are unlikely).

We agree with the reviewer and have modified all instances of Glossina to G. morsitans.

- Lines 258-259: See my comment above on the convenience of a brief explanation of the position of Glossinidae flies within the Hippoboscoidea and the common reproductive features. Many potential readers might be unaware of that so a simple clarification would not bother.

Thank you for this comment. We have revised the text in this section to more clearly describe the relationship between the membership of the Hippoboscoidea. This should provide clarity to readers not familiar with this group and their relationships. Lines 262-269

“The other families within the Hippoboscoidea superfamily (Hippoboscidae (Ked flies), Nycteribiidae (Bat flies) and Streblidae (Bat flies)) are all obligately hematophagous and undergo obligate viviparity. This suggests that members of these families likely have similar reproductive adaptations, but may also have species or family specific specializations.  Comparative analysis of G. morsitanslossina uterine morphology with that of other viviparous flies from the Hippoboscoidea superfamily (keds and bat flies) and with that of oviparous members of the brachyceran Diptera, such as Drosophila anda members of the Oestroidea and Muscoidea superfamilies and Musca, can inform as to the derivation of these adaptations.”

- Line 260: A comparison with Drosophila would be of course interesting and a logical option given the model organism status of fruit flies, but perhaps it would be more interesting to compare with more closely related calyptrate flies (Musca but also other Muscoidea and Oestroidea flies).

See reply to the previous comment.

- Lines 472-475: The unparalleled opportunities provided by microCT to morphological studies are enormous and this paper is a very good example, but I like that the authors also emphasize the specific opportunities for medical and veterinary entomology studies. In this sense, a potentially interesting microCT study related to tsetse flies reproduction and development could be describing the intrauterine larval development using this non-invasive technique. A very recent study has described the morphology of the excretory and digestive systems through the larval instars of bot flies using microCT (Major differences in the larval anatomy of the digestive and excretory systems of three Oestridae species revealed by micro‐CT; https://doi.org/10.1111/mve.12476), and visualizing the development of those systems would be particularly interesting in the larvae of tsetse flies.

Thank you for alerting us to this study! We have added the following lines to the conclusion and a reference to the paper. Lines 473-480.

“Recent comparative micro CT based work was performed comparing the larval digestive tract across three larval instars in the sheep nasal bot fly species (Oestrus ovis L.)). In addition, they made comparisons with two other species Cephenemyia stimulator (Clark) and Hypoderma actaeon (Brauer) This study discovered significant differences in the morphology of the digestive tract over instars and between species. highlighting the potential for comparative analysis using this technique [45]. The authors there noted that the time investment required to perform a detailed segmentation of scanned volumes is a limiting factor.”

Reviewer 2 Report

A fascinating paper, and technique! Only some very minor corrections as follows.

All text: depending on the journal format, set Latin genus and species names in italics.

Line 14: correct „They only...“

Line 32: insert „with“ to „...associated with...“

Line 46: insert „in“ to: „...and Nagana in animals“

Line 181: delete „the“ in „...reveals how tightly...“

References: revise formatting, such as Glossina-Morsitans-Morsitans which should read Glossina morsitans morsitans (all in italics if applicable)

Author Response

All text: depending on the journal format, set Latin genus and species names in italics.

Corrected

Line 14: correct „They only…“

Corrected

Line 32: insert „with“ to „...associated with...“

Corrected

Line 46: insert „in“ to: „...and Nagana in animals“

Corrected

 Line 181: delete „the“ in „...reveals how tightly...“

Corrected

References: revise formatting, such as Glossina-Morsitans-Morsitans which should read Glossina morsitans morsitans (all in italics if applicable)

Thank you for catching this all references have been revised with proper italicization.